# Revisiting Consistency for Semi-Supervised Semantic Segmentation [note 1]

**DOI:** 10.3390/s23020940

**Published:** 2023-01-13

**Authors:** Ivan Grubišić, Marin Oršić, Siniša Šegvić

**Affiliations:** 1Faculty of Electrical Engineering and Computing, University of Zagreb, Unska 3, 10000 Zagreb, Croatia; 2Microblink Ltd., Strojarska Cesta 20, 10000 Zagreb, Croatia

**Keywords:** semi-supervised learning, semantic segmentation, dense prediction, one-way consistency, deep learning, scene understanding

## Abstract

Semi-supervised learning is an attractive technique in practical deployments of deep models since it relaxes the dependence on labeled data. It is especially important in the scope of dense prediction because pixel-level annotation requires substantial effort. This paper considers semi-supervised algorithms that enforce consistent predictions over perturbed unlabeled inputs. We study the advantages of perturbing only one of the two model instances and preventing the backward pass through the unperturbed instance. We also propose a competitive perturbation model as a composition of geometric warp and photometric jittering. We experiment with efficient models due to their importance for real-time and low-power applications. Our experiments show clear advantages of (1) one-way consistency, (2) perturbing only the student branch, and (3) strong photometric and geometric perturbations. Our perturbation model outperforms recent work and most of the contribution comes from the photometric component. Experiments with additional data from the large coarsely annotated subset of Cityscapes suggest that semi-supervised training can outperform supervised training with coarse labels. Our source code is available at https://github.com/Ivan1248/semisup-seg-efficient.

## 1. Introduction

Most machine learning applications are hampered by the need to collect large annotated datasets. Learning with incomplete supervision [1,2] presents a great opportunity to speed up the development cycle and enable rapid adaptation to new environments. Semi-supervised learning [3,4,5] is especially relevant in the dense prediction context [6,7,8] since pixel-level labels are very expensive, whereas unlabeled images are easily obtained.

Dense prediction typically operates on high resolutions in order to be able to recognize small objects. Furthermore, competitive performance requires learning on large batches and large crops [9,10,11]. This typically entails a large memory footprint during training, which constrains model capacity [12]. Many semi-supervised algorithms introduce additional components to the training setup. For instance, training with surrogate classes [13] implies infeasible logit tensor size, while GAN-based approaches require an additional generator [6,14] or discriminator [7,15,16]. Some other approaches require multiple model instances [17,18,19,20] or accumulated predictions across the dataset [21]. Such designs are less appropriate for dense prediction since they constrain model capacity.

This paper studies semi-supervised approaches [3,5,18,21,22] that require consistent predictions over input perturbations. In the considered consistency objective, input perturbations affect only one of two model instances, while the gradient is not propagated towards the model instance which operates on the clean (weakly perturbed) input [4,5]. For brevity, we refer to the two model instances as the perturbed branch and the clean branch. If the gradient is not computed in a branch, we refer to it as the teacher, and otherwise as the student. Hence, we refer to the considered approach as a one-way consistency with the clean teacher.

Let x be the input, *T* a perturbation to which the ideal model should be invariant, $h_{\theta }$ the student, and $h_{\theta^{\prime }}$ the teacher, where $\theta^{\prime }$ denotes a frozen copy of the student parameters θ. Then, one-way consistency with clean teacher can be expressed as a divergence *D* between the two predictions:(1)$$L_{\theta }^{\mathrm{ct}}\left(x,T\right)=D\left(h_{\theta^{\prime }}\left(x\right),h_{\theta }\left(T\left(x\right)\right)\right)\;.$$

We argue that the clean teacher approach is a method of choice in case of perturbations that are too strong for standard data augmentation. In this setting, perturbed inputs typically give rise to less reliable predictions than their clean counterparts. Figure 1 illustrates the advantage of the clean teacher approach in comparison with other kinds of consistency on the Two moons dataset. The clean student experiment (Figure 1b) shows that many blue data points get classified into the red class due to teacher inputs being pushed towards labeled examples of the opposite class. This aberration does not occur when the teacher inputs are clean (Figure 1c). Two-way consistency [21] (Figure 1d) can be viewed as a superposition of the two one-way approaches and works better than (Figure 1b), but worse than (Figure 1c). In our experiments, *D* corresponds to KL divergence.

One-way consistency is especially advantageous in the dense prediction context since it does not require caching latent activations in the teacher. This allows for better training in many practical cases where model capacity is limited by GPU memory [12,23]. In comparison with two-way consistency [20,21], the proposed approach both improves generalization and approximately halves the training memory footprint.

This paper is an extended version of our preliminary conference report [24]. It exposes the elements of our method in much more detail and complements them with many new experiments. In particular, the most important additions are additional ablation and validation studies, full-resolution Cityscapes experiments, and a detailed analysis of a large-scale experiment that compares the contribution of coarse labels with semi-supervised learning on unlabeled images. The new experiments add more evidence in favor of one-way consistency with respect to other consistency variants, investigate the influence of particular components of our algorithm and various hyper-parameters, and investigate the behavior of the proposed algorithm in different data regimes (higher resolution; additional unlabeled images).

The consolidated paper proposes a simple and effective method for semi-supervised semantic segmentation. One-way consistency with clean teacher [4,5,25] outperforms the two-way formulation in our validation experiments. In addition, it retains the memory footprint of supervised training because the teacher activations depend on parameters that are treated as constants. Experiments with a standard convolutional architecture [26] reveal that our photometric and geometric perturbations lead to competitive generalization performance and outperform their counterpart from a recent related work [25]. A similar advantage can be observed in experiments with a recent efficient architecture [27], which offers a similar performance while requiring an order of magnitude of less computation. To our knowledge, this is the first account of the evaluation of semi-supervised algorithms for dense prediction with a model capable of real-time inference. This contributes to the goals of Green AI [28] by enabling competitive research with less environmental damage.

This paper proceeds as follows. Section 2 presents related work. Section 3 describes the one-way consistency objective adapted to dense prediction, our perturbation model, and a description of our memory-efficient consistency training procedure. Section 4 presents the experimental setup, which includes information about datasets and training details, as well as the performed experiments in semi-supervised semantic segmentation. Finally, Section 5 presents the conclusion.

## 2. Related Work

Our work spans the fields of dense prediction and semi-supervised learning. The proposed methodology is most related to previous work in semi-supervised semantic segmentation.

### 2.1. Dense Prediction

Image-wide classification models usually achieve efficiency, spatial invariance, and integration of contextual information by gradual downsampling of representations and use of global spatial pooling operations. However, dense prediction also requires location accuracy. This emphasizes the trade-off between efficiency and quality of high-resolution features in the model design. Some common designs use a classification backbone as a feature encoder and attach a decoder that restores the spatial resolution. Many approaches seek to enhance contextual information, starting with FCN-8s [29]. UNet [30] improves spatial details by directly using earlier representations of the encoder in a symmetric decoder. Further work improves the efficiency with lighter decoders [23,31]. Some models use context aggregation modules such as spatial pyramid pooling [32] and multi-scale inference [31,33]. DeepLab [26] increases the receptive field through dilated convolutions and improves spatial details through CRF post-processing. HRNet [34] maintains the full resolution throughout the whole model and incrementally introduces parallel lower-resolution branches that exchange information between stages. Semantic segmentation gains much from ImageNet pre-trained encoders [23,26].

### 2.2. Semi-Supervised Learning

Semi-supervised methods often rely on some of the following assumptions about the data distribution [35]: (1) similar inputs in high density regions correspond to similar outputs (smoothness assumption), (2) inputs form clusters separated by low-density regions and inputs within clusters are likely to correspond to similar outputs (cluster assumption), and (3) the data lies on a lower-dimensional manifold (manifold assumption). Semi-supervised methods devise various inductive biases that exploit such regularities for learning from unlabeled data.

Entropy minimization [36] encourages high confidence in unlabeled inputs. Such designs push decision boundaries towards low-density regions, under assumptions of clustered data and prediction smoothness. Pseudo-label training (or self-training) [37,38,39] also encourages high confidence (because of hard pseudo-labels) as well as consistency with a previously trained teacher. The basic forms of such algorithms do not achieve competitive performance on their own [40], but can be effective in conjunction with other approaches [4,41]. Pseudo-labels can be made very effective by confidence-based selection and other processing [37,38,42]. Note that some concurrent work [42] uses the term pseudo-label as a synonym for processed teacher prediction in one-way consistency, but we do not follow this practice.

Many approaches exploit the smoothness assumption by enforcing prediction consistency across different versions of the same input or different model instances. Introducing knowledge about equivariance has been studied for understanding and learning useful image representations [43,44] and improving dense prediction [45,46,47]. Exemplar training [13] associates patches with their original images (each image is a separate surrogate class). Temporal ensembling [21] enforces per-datapoint consistency between the current prediction and a moving average of past predictions. Mean Teacher [3] encourages consistency with a teacher whose parameters are an exponential moving average of the student’s parameters.

Clusterization of latent representations can be promoted by penalizing walks which start in a labeled example, pass over an unlabeled example, and end in another example with a different label [48]. PiCIE [46] obtains semantically meaningful segmentation without labels by jointly learning clustering and representation consistency under photometric and geometric perturbations.

MixMatch [49] encourages consistency between predictions in different MixUp perturbations of the same input. The average prediction is used as a pseudo-label for all variants of the input. Deep co-training [19] produces complementary models by encouraging them to be consistent while each is trained on adversarial examples of the other one.

Consistency losses may encourage trivial solutions, where all inputs give rise to the same output. This is not much of a problem in semi-supervised learning since there the trivial solution is inhibited through the supervised objective. Interestingly, recent work shows that a variant of simple one-way consistency evades trivial solutions even in the context of self-supervised representation learning [50,51].

Virtual adversarial training (VAT) [4] encourages one-way consistency between predictions in original datapoints and their adversarial perturbations. These perturbations are recovered by maximizing a quadratic approximation of the prediction divergence in a small $L^{2}$ ball around the input. Better performance is often obtained by additionally encouraging low-entropy predictions [36]. Unsupervised data augmentation (UDA) [5] also uses a one-way consistency loss. FixMatch [52] shows that pseudo-label selection and processing can be useful in a one-way consistency. However, instead of adversarial additive perturbations, they use random augmentations generated by RandAugment. Different from all previous approaches, we explore an exhaustive set of consistency formulations.

### 2.3. Semi-Supervised Semantic Segmentation

In the classic semi-supervised GAN (SGAN) setup, the classifier also acts as a discriminator which distinguishes between real data (both labeled and unlabeled) and fake data produced by the generator [14]. This approach has been adapted for dense prediction by expressing the discriminator as a segmentation network that produces dense C+1-way logits [6]. KE-GAN [53] additionally enforces semantic consistency of neighbouring predictions by leveraging label-similarity recovered from a large text corpus (MIT ConceptNet). A semantic segmentation model can also be trained as a GAN generator (AdvSemSeg) [7]. In this setup, the discriminator guesses whether its input is ground truth or generated by the segmentation network. The discriminator is also used to choose better predictions for use as pseudo-labels for semi-supervised training. s4GAN + MLMT [8] additionally post-processes the recovered dense predictions by emphasizing classes identified by an image-wide classifier trained with Mean Teacher [3]. The authors note that the image-wide classification component is not appropriate for datasets such as Cityscapes, where almost all images contain a large number of classes.

A recent approach enforces consistency between outputs of redundant decoders with noisy intermediate representations [54]. Other recent work studies pseudo-labeling in the dense prediction context [55,56,57]. Zhu et al. [55] observe advantages in hard pseudo-labels. A recent approach [20] proposes a two-way consistency loss, which is related to the Π-model [21], and perturbs both inputs with geometric warps. However, we show that perturbing only the student branch generalizes better and has a smaller training footprint. A concurrent work [58] successfully applies a contrastive loss [59,60] between two branches which receive overlapping crops, and proposes a pixel-dependent consistency direction. Mean Teacher consistency with CutMix perturbations achieved state-of-the-art performance on half-resolution Cityscapes [25] prior to this work. Different than most presented approaches and similar to [25,55,56,61], our method does not increase the training footprint [12]. In comparison with [55,56,61], our teacher is updated in each training step, which eliminates the need for multiple training episodes. In comparison with [25], this work proposes a perturbation model which results in better generalization and shows that simple one-way consistency can be competitive with Mean Teacher. None of the previous approaches addresses semi-supervised training of efficient dense prediction models. We examine the simplest forms of consistency, explain advantages of perturbing only the student with respect to other forms of consistency, and propose a novel perturbation model. None of the previous approaches considered semi-supervised training of efficient dense-prediction models, nor studied composite perturbations of photometry and geometry.

## 3. Method

We formulate dense consistency as a mean pixel-wise divergence between corresponding predictions in the clean image and its perturbed version. We perturb images with a composition of photometric and geometric transformations. Photometric transformations do not disturb the spatial layout of the input image. Geometric transformations affect the spatial layout of the input image and the same kind of disturbance is expected at the model output. Ideally, our models should exhibit invariance to photometric transformations and equivariance [44] to the geometric ones.

### 3.1. Notation

We typeset vectors and arrays in bold, sets in blackboard bold, and we underline random variables. $P\left[\underline{y}|\underline{x}=x\right]$ denotes the distribution of a random variable $\underline{y}|x$, while P(y|x) is a shorthand for the probability $P(\underline{y}=y|\underline{x}=x)$. We denote the expectation of a function of a random variable as e.g., $E_{\underline{\tau }}f\left(\tau \right)$. We use similar notation to denote the average over a set: $E_{x\in D}f\left(x\right)$. We use the Iverson bracket notation: given a statement *P*, ⟦P⟧=1 if *P* is true; 0 otherwise. We denote cross-entropy with $H_{\underline{y}}({\underline{y}}^{*}):=E_{y∼{\underline{y}}^{*}}\ln p\left(\underline{y}=y\right)$, and entropy with $H(\underline{y})$ [62]. We use Python-like array indexing notation.

We denote the labeled dataset as $D_{l}$, and the unlabeled dataset as $D_{u}$. We consider input images $x\in {\left[0,1\right]}^{H\times W\times 3}$ and dense labels $y\in {\left\{1\;\ldots \;C\right\}}^{H\times W}$. A model instance maps an image to per-pixel class probabilities: $h_{\theta }{\left(x\right)}_{\left[i,j,c\right]}=P\left({\underline{y}}_{\left[i,j\right]}=c|x,\theta \right)$. For convenience, we identify output vectors of class probabilities with distributions: $h_{\theta }{\left(x\right)}_{\left[i,j\right]}\equiv P\left[{\underline{y}}_{\left[i,j\right]}|x,\theta \right]$.

### 3.2. Dense One-Way Consistency

We adapt one-way consistency [4,5] for dense prediction under our perturbation model $T_{\tau }=T_{\gamma }^{G}\circ T_{\varphi }^{P}$, where $T_{\gamma }^{G}$ is a geometric warp, $T_{\varphi }^{P}$ a per-pixel photometric perturbation, and τ=(γ,φ) perturbation parameters. $T_{\gamma }^{G}$ displaces pixels with respect to a dense deformation field. The same geometric warp is applied to the student input and the teacher output. Figure 2 illustrates the computational graph of the resulting dense consistency loss. In simple one-way consistency, the teacher parameters $\theta^{\prime }$ are a frozen copy of the student parameter θ. In Mean Teacher, $\theta^{\prime }$ is a moving average of θ. In simple two-way consistency, both branches use the same θ and are subject to gradient propagation.

A general semi-supervised training criterion $L(\theta ;D_{l},D_{u})$ can be expressed as a weighted sum of a supervised term $L_{s}$ over labeled data and an unsupervised consistency term $L_{c}$: (2)$$L\left(\theta ;D_{l},D_{u}\right)=\underset{\left(x,y\right)\in D_{l}}{E}L_{s}\left(\theta ;x,y\right)+\alpha \underset{x\in D_{u}}{E}\underset{\underline{\tau }}{E}L_{c}\left(\theta ;x,\tau \right).$$
In our experiments, $L_{s}$ is the usual mean per-pixel cross entropy with $L^{2}$ regularization. We stochastically estimate the expectation over perturbation parameters $\underline{\tau }$ with one sample per training step.

We formulate the unsupervised term $L_{c}$ at pixel (i,j) as a one-way divergence *D* between the prediction in the perturbed image and its interpolated correspondence in the clean image. The proposed loss encourages the trained model to be equivariant to $T_{\gamma }^{G}$ and invariant to $T_{\varphi }^{P}$:(3)$$\begin{array}{c} L_{c}^{i,j}\left(\theta ;x,\tau \right)=D\left(T_{\gamma }^{G}{\left(h_{\theta^{\prime }}\left(x\right)\right)}_{\left[i,j\right]},h_{\theta }{\left(\left(T_{\gamma }^{G}\circ T_{\varphi }^{P}\right)\left(x\right)\right)}_{\left[i,j\right]}\right)\;. \end{array}$$
We use a validity mask $v^{\gamma }\in {\left\{0,1\right\}}^{H\times W}$, $v_{\left[i,j\right]}^{\gamma }=⟦T_{\gamma }^{G}{\left(1_{H\times W}\right)}_{\left[i,j\right]}=1⟧$ to ensure that the loss is unaffected by padding sampled from outside of $\left[1,H\right]\times \left[1,W\right]$. A vector produced by $T_{\gamma }^{G}{\left(h_{\theta }\left(x\right)\right)}_{\left[i,j\right]}$ represents a valid distribution wherever $v_{\left[i,j\right]}^{\gamma }=1$. Finally, we express the consistency term $L_{c}$ as a mean contribution over all pixels:(4)$$L_{c}\left(\theta ;x,\tau \right)=\frac{1}{\sum (v^{\gamma })}\sum_{i,j}v_{\left[i,j\right]}^{\gamma }L_{c}^{i,j}\left(\theta ;x,\tau \right)\;.$$

Recall that the gradient is not computed with respect to $\theta^{\prime }$. Consequently, $L_{c}$ allows gradient propagation only towards the perturbed image. We refer to such training as one-way consistency with clean teacher (and perturbed student). Such formulation provides two distinct advantages over other kinds of consistency. First, predictions in perturbed images are pulled towards predictions in clean images. This improves generalization when the perturbations are stronger than data augmentations used in $L_{s}$ (cf. Figure 1 and Figure 3). Second, we do not have to cache teacher activations during training since the gradients propagate only towards the student branch. Hence, the proposed semi-supervised objective does not constrain model complexity with respect to the supervised baseline.

We use KL divergence as a principled choice for D: (5)$$D\left(\underline{y},\overset{\sim }{\underset{\_}{y}}\right):=\underset{\underline{y}}{E}\ln\frac{P(\underline{y}=y)}{P(\overset{\sim }{\underset{\_}{y}}=y)}=H_{\overset{\sim }{\underset{\_}{y}}}\left(\underline{y}\right)-H\left(\underline{y}\right)\;.$$
Note that the entropy term $-H(\underline{y})$ does not affect parameter updates since the gradients are not propagated through $\theta^{\prime }$. Hence, one-way consistency does not encourage increasing entropy of model predictions in clean images. Several researchers have observed improvement after adding an entropy minimization term [36] to the consistency loss [4,5]. This practice did not prove beneficial in our initial experiments.

Note that two-way consistency [20,21] would be obtained by replacing $\theta^{\prime }$ with θ. It would require caching latent activations for both model instances, which approximately doubles the training footprint with respect to the supervised baseline. This would be undesirable due to constraining the feasible capacity of the deployed models [12,63].

We argue that consistency with clean teacher generalizes better than consistency with clean student since strong perturbations may push inputs beyond the natural manifold and spoil predictions (cf. Figure 1). Moreover, perturbing both branches sometimes results in learning to map all perturbed pixels to similar arbitrary predictions (e.g., always the same class) [64]. Figure 3 illustrates that consistency training has the best chance to succeed if the teacher is applied to the clean image, and the student learns on the perturbed image.

### 3.3. Photometric Component of the Proposed Perturbation Model

We express pixel-level photometric transformations as a composition of five simple perturbations with five image-wide parameters φ=(b,s,h,c,π). These perturbations are applied in each pixel in the following order: (1) brightness is shifted by adding *b* to all channels, (2) saturation is multiplied with *s*, (3) hue is shifted by addition with *h*, (4) contrast is modulated by multiplying all channels with *c*, and (5) RGB channels are randomly permuted according to π. The resulting compound transformation $T_{\varphi }^{P}$ is independently applied to all image pixels.

Our training procedure randomly picks image-wide parameters φ for each unlabeled image. The parameters are sampled as follows: b∼U(−0.25,0.25), s∼U(0.25,2), $h∼U(-36^{\circ },36^{\circ })$, c∼U(0.25,2), and $\pi ∼U(S_{3})$, where $S_{3}$ represents the set of all 6 3-element permutations.

### 3.4. Geometric Component of the Proposed Perturbation Model

We formulate a fairly general class of parametric geometric transformations by leveraging thin plate splines (TPS) [65,66]. We consider the 2D TPS warp $f:R^{2}\to R^{2}$, which maps each image coordinate pair q to the relative 2D displacement of its correspondence $q^{\prime }$:(6)$$f\left(q\right)=q^{\prime }-q\;.$$
TPS warps minimize the bending energy (curvature) $\int_{\mathrm{dom}(f)}{∥\frac{\partial^{2}f\left(q\right)}{\partial q^{2}}∥}_{F}^{2}dq$ given a set of control points and their displacements $\left\{(c_{i},d_{i}):i=1\;\ldots \;n\right\}\subset R^{2}\times R^{2}$. In simple words, a TPS warp produces a smooth deformation field which optimally satisfies all constraints $f\left(c_{i}\right)=d_{i}$. In the 2D case, the solution of the TPS problem takes the following form:(7)$$f\left(q\right)=A\left[\begin{array}{c} 1\\ q \end{array}\right]+W{\left[\begin{array}{c} \phi (∥q-c_{i}∥) \end{array}\right]}_{i=1\;\ldots \;n}^{T}\;,$$
where q denotes a 2D coordinate vector to be transformed, A is a 2×3 affine transformation matrix, W is a 2×n control point coefficient matrix, and $\phi \left(r\right)=r^{2}\ln\left(r\right)$. Such a 2D TPS warp is equivariant to rotation and translation [66]. That is, f(T(q))=T(f(q)) for every composition of rotation and translation *T*.

TPS parameters A and W can be determined as a solution of a standard linear system which enforces deformation constraints $\left(c_{i},d_{i}\right)$, and square-integrability of second derivatives of *f*. When we determine A and W, we can easily transform entire images.

We first consider images as continuous domain functions and later return to images as arrays from ${\left[0,1\right]}^{H\times W\times 3}$. Let $I:\mathrm{dom}\left(I\right)\to {\left[0,1\right]}^{3}$ be the original image of size (W,H), where $\mathrm{dom}(I)=\left[0,W\right]\times \left[0,H\right]$. Then the transformed image $I^{\prime }$ can be expressed as
(8)$$I^{\prime }\left(q+f\left(q\right)\right)=\left\{\begin{array}{cc} I(q), & q\in \mathrm{dom}(I),\\ 0, & \mathrm{otherwise}. \end{array}\right.$$

The resulting formulation is known as forward warping [67] and is tricky to implement. We, therefore, prefer to recover the reverse transformation ff˜, which can be conducted by replacing each control point $c_{i}$ with $c_{i}^{\prime }=c_{i}+d_{i}$. Then, the transformed image is:(9)II˜(q′)=I(q′−ff˜(q′)),q′−ff˜(q′)∈dom(I),0,otherwise.
This formulation is known as backward warping [67]. It can be easily implemented for discrete images by leveraging bilinear interpolation. Contemporary frameworks already include the implementations for the GPU hardware. Hence, the main difficulty is to determine the TPS parameters by solving two linear systems with (n+3)×(n+3) variables [66].

In our experiments, we use n=4 control points corresponding to the centers of the four image quadrants: $\left(c_{1}^{\prime },\ldots ,c_{4}^{\prime }\right)=\left({\left[\frac{1}{4}H,\frac{1}{4}W\right]}^{T},\ldots ,{\left[\frac{3}{4}H,\frac{3}{4}W\right]}^{T}\right)$. The parameters of our geometric transformation are four 2D displacements $\gamma =\left(d_{1},\ldots ,d_{4}\right)$. Let $f_{\gamma }$ denote the resulting TPS warp. Then, we can express our transformation as $T_{\gamma }^{G}\left(x\right)=\mathrm{backward}\_\mathrm{warp}\left(x,f_{\gamma }\right)$.

Our training procedure picks a random γ for each unlabeled image. Each displacement is sampled from a bivariate normal distribution $N\left(0_{2},0.05\times H\times I_{2}\right)$, where *H* is the height of training crops.

### 3.5. Training Procedure

Algorithm 1 sketches a procedure for recovering gradients of the proposed semi-supervised loss (Equation 2) on a mixed batch of labeled and unlabeled examples. For simplicity, we make the following changes in notation here: $x_{l}$ and $y_{l}$ are batches of size $B_{l}$, $x_{u}$, γ and φ batches of size $B_{u}$, and all functions are applied to batches. The algorithm computes the gradient of the supervised loss, discards cached activations, computes the teacher predictions, applies the consistency loss (Equation 3), and finally accumulates the gradient contributions of the two losses. Backpropagation through one-way consistency with clean teacher requires roughly the same extent of caching as in the supervised baseline. Hence, our approach constrains the model complexity much less than the two-way consistency.
**Algorithm 1.** Evaluation of the gradient of the proposed semi-supervised loss given perturbationparameters (γ, φ) on a mixed batch of labeled ($x_{l}$, $y_{l}$) and unlabeled ($x_{u}$) examples. CE denotesmean cross entropy, while KL_masked denotes mean KL divergence over valid pixels.
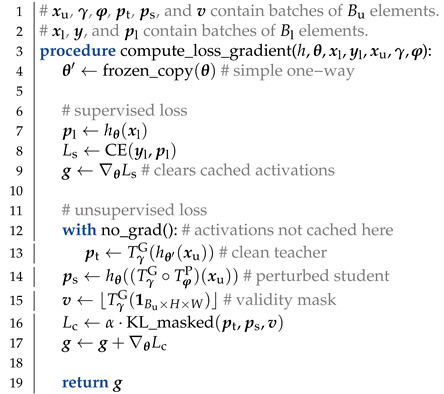


Figure 4 illustrates GPU memory allocation during a semi-supervised training iteration of a SwiftNet-RN34 model with one-way and two-way consistency. We recovered these measurements by leveraging the following functions of the torch.cuda package: max_memory_allocated, memory_allocated, reset_peak_memory_stats, and empty_cache. The training was carried out on a RTX A4500 GPU. Numbers on the x-axis correspond to lines of the pseudo-code in Algorithm 1. Line 9 backpropagates through the supervised loss and caches the gradients. The memory footprint briefly peaks due to temporary storage and immediately declines since PyTorch automatically releases all cached activations immediately after the backpropagation. Line 13 computes the teacher output. This step does not cache intermediate activations due to torch.no_grad. Line 16 computes the unsupervised loss, which requires the caching of activations on a large spatial resolution. The memory footprint briefly peaks since we delete perturbed inputs and teacher predictions immediately after line 16 (for simplicity, we omit opportunistic deletions from Algorithm 1). Line 17 triggers the backpropagation algorithm and accumulates the gradients of the consistency loss. The memory footprint briefly peaks due to temporary storage and immediately declines due to automatic deletion of the cached activations. At this point, the memory footprint is slightly greater than at line 4 since we still hold the supervised predictions in order to accumulate the recognition performance on the training dataset.

The ratio between memory allocations at lines 16 and 9 reveals the relative memory overhead of our semi-supervised approach. Note that the absolute overhead is model independent since it corresponds to the total size of perturbed inputs and predictions, and intermediate results of dense KL-divergence. On the other hand, the memory footprint of the supervised baseline is model dependent, since it reflects the computational complexity of the backbone. Consequently, the relative overhead approaches 1 as the model size increases, and is around 1.26 for SwiftNet-RN34.

## 4. Results

Our experiments evaluate one-way consistency with clean teacher and a composition of photometric and geometric perturbations ($T_{\gamma }^{G}\circ T_{\phi }^{P}$). We compare our approach with other kinds of consistency and the state of the art in semi-supervised semantic segmentation. We denote simple one-way consistency as “simple”, Mean Teacher [3] as “MT”, and our perturbations as “PhTPS”. In experiments that compare consistency variants, “1w” denotes one-way, “2w” denotes two way, “ct” denotes clean teacher, “cs” denotes clean student, and “2p” denotes both inputs perturbed. We present semi-supervised experiments in several semantic segmentation setups as well as in image-classification setups on CIFAR-10. Our implementations are based on the PyTorch framework [68].

### 4.1. Experimental Setup

Datasets.We perform semantic segmentation on Cityscapes [9], and image classification on CIFAR-10. Cityscapes contains 2975 training, 500 validation and 1525 testing images with resolution 1024×2048. Images are acquired from a moving vehicle during daytime and fine weather conditions. We present half-resolution and full-resolution experiments. We use bilinear interpolation for images and nearest neighbour subsampling for labels. Some experiments on Cityscapes also use the coarsely labeled Cityscapes subset (“train-extra”) that contains 19,998 images. CIFAR-10 consists of 50,000 training and 10,000 test images of resolution 32×32.

Common setup. We include both unlabeled and labeled images in $D_{u}$, which we use for the consistency loss. We train on batches of $B_{l}$ labeled and $B_{u}$ unlabeled images. We perform $\left\lfloor \left|D_{l}\right|/B_{l}\right\rfloor$ training steps per epoch. We use the same perturbation model across all datasets and tasks (TPS displacements are proportional to image size), which is likely suboptimal [69]. Batch normalization statistics are updated only in non-teacher model instances with clean inputs except for full-resolution Cityscapes, for which updating the statistics in the perturbed student performed better in our validation experiments (cf. Appendix B). The teacher always uses the estimated population statistics, and does not update them. In Mean Teacher, the teacher uses an exponential moving average of the student’s estimated population statistics.

Semantic segmentation setup. Cityscapes experiments involve the following models: SwiftNet with ResNet-18 (SwiftNet-RN18) or ResNet-34 (SwiftNet-RN34), and DeepLab v2 with a ResNet-101 backbone. We initialize the backbones with ImageNet pre-trained parameters. We apply random scaling, cropping, and horizontal flipping to all inputs and segmentation labels. We refer to such examples as clean. We schedule the learning rate according to e↦ηcos(eπ/2), where $e\in \left[0\ldots 1\right]$ is the fraction of epochs completed. This alleviates the generalization drop at the end of training with standard cosine annealing [70]. We use learning rates $\eta =4\times 10^{-4}$ for randomly initialized parameters and $\eta =10^{-4}$ for pre-trained parameters. We use Adam with $\left(\beta_{1},\beta_{2}\right)=\left(0.9,0.99\right)$. The L${}^{2}$ regularization weight in supervised experiments is $10^{-4}$ for randomly initialized and $2.5\times 10^{-5}$ for pre-trained parameters [27]. We have found that such L${}^{2}$ regularization is too strong for our full-resolution semi-supervised experiments. Thus, we use a 4× smaller weight there. Based on early validation experiments, we use α=0.5 unless stated otherwise. Batch sizes are $\left(B_{l},B_{u}\right)=\left(8,8\right)$ for SwiftNet-RN18 [27] and $\left(B_{l},B_{u}\right)=\left(4,4\right)$ for DeepLab v2 (ResNet-101 backbone) [26]. The batch size in corresponding supervised experiments is $B_{l}$.

In half-resolution Cityscapes experiments the size of crops is 448×448 and the logarithm of the scaling factor is sampled from $U(\ln\left(1.5^{-1}\right),\ln\left(1.5\right))$. We train SwiftNet for $200\times \frac{2975}{\left|D_{l}\right|}$ epochs (200 epochs or 74,200 iterations when all labels are used), and DeepLab v2 for $100\times \frac{2975}{\left|D_{l}\right|}$ epochs (100 epochs or 74,300 iterations when all labels are used). In comparison with SwiftNet-RN18, DeepLab v2 incurs a 12-fold per-image slowdown during supervised training. However, it also requires less epochs since it has very few parameters with random initialization. Hence, semi-supervised DeepLab v2 trains more than 4× slower than SwiftNet-RN18 on RTX 2080Ti. Section A.2 presents more detailed comparisons of memory and time requirements of different semi-supervised algorithms.

Our full-resolution experiments only use SwiftNet models. The crop size is 768×768 and the spatial scaling is sampled from $U(2^{-1},2)$. The number of epochs is 250 when all labels are used. The batch size is 8 in supervised experiments, and $\left(B_{l},B_{u}\right)=\left(8,8\right)$ in semi-supervised experiments.

Section A.1 presents an overview and comparison of hyper-parameters with other consistency-based methods that are compared in the experiments.

Classification setup. Classification experiments target CIFAR-10 and involve the Wide ResNet model WRN-28-2 with standard hyper-parameters [71]. We augment all training images with random flips, padding and random cropping. We use all training images (including labeled images) in $D_{u}$ for the consistency loss. Batch sizes are $\left(B_{l},B_{u}\right)=\left(128,640\right)$. Thus, the number of iterations per epoch is $\left\lfloor \frac{\left|D_{l}\right|}{128}\right\rfloor$. For example, only one iteration is performed if $\left|D_{l}\right|=250$. We run $1000\times \frac{4000}{\left|D_{l}\right|}$ epochs in semi-supervised, and 100 epochs in supervised training. We use default VAT hyper-parameters $\xi =10^{-6}$, ϵ=10, α=1 [4]. We perform photometric perturbations as described, and sample TPS displacements from $N(0,3.2\times I_{2})$.

Evaluation. We report generalization performance at the end of training. We report sample means and sample standard deviations (with Bessel’s correction) of the corresponding evaluation metric (mIoU or classification accuracy) of 5 training runs, evaluated on the corresponding validation dataset.

### 4.2. Semantic Segmentation on Half-Resolution Cityscapes

Table 1 compares our approach with the previous state of the art. We train using different proportions of training labels and evaluate mIoU on half-resolution Cityscapes val. The top section presents the previous work [7,8,25,57]. The middle section presents our experiments based on DeepLab v2 [26]. Note that here we outperform some previous work due to more involved training (as described in Section 4.1). since that would be a method of choice in all practical applications. Hence, we get consistently greater performance. We perform a proper comparison with [25] by using our training setup in combination with their method. Our MT-PhTPS outperforms MT-CutMix with L2 loss and confidence thresholding when 1/4 or more labels are available, while underperforming with 1/8 labels.

The bottom section involves the efficient model SwiftNet-RN18. Our perturbation model outperforms CutMix both with simple consistency, as well as with Mean Teacher. Overall, Mean Teacher outperforms simple consistency. We observe that DeepLab v2 and SwiftNet-RN18 get very similar benefits from the consistency loss. SwiftNet-RN18 comes out as a method of choice due to about 12× faster inference than DeepLab v2 with ResNet-101 on RTX 2080Ti (see Section A.2 for more details). Experiments from the middle and the bottom section use the same splits to ensure a fair comparison.

Now, we present ablation and hyper-parameter validation studies for simple-PhTPS consistency with SwiftNet-RN18. Table 2 presents ablations of the perturbation model, and also includes supervised training with PhTPS augmentations in one half of each mini-batch in addition to standard jittering. Perturbing the whole mini-batch with PhTPS in supervised training did not improve upon the baseline. We observe that perturbing half of each mini-batch with PhTPS in addition to standard jittering improves the supervised performance, but quite less than semi-supervised training. Semi-supervised experiments suggest that photometric perturbations (Ph) contribute most, and that geometric perturbations (TPS) are not useful when there is 1/2 or more of the labels.

Figure 5 shows perturbation strength validation using 1/4 of the labels. Rows correspond to the factor that multiplies the standard deviation of control point displacements $s_{G}$ defined at the end of Section 3.4. Columns correspond to the strength of the photometric perturbation $s_{P}$. The photometric strength $s_{P}$ modulates the random photometric parameters according to the following expression: (10)$$\begin{array}{c} \left(\underline{b},\underline{s},\underline{h},\underline{c}\right)\mapsto (s_{P}\times \underline{b},\exp\left(s_{P}\times \ln\underline{s}\right),s_{P}\times \underline{h},\exp\left(s_{P}\times \ln\left(\underline{c}\right)\right)\;. \end{array}$$

We set the probability of choosing a random channel permutation as $\min\{s_{P},1\}$. Hence, $s_{P}=0$ corresponds to the identity function. Note that the “1/4” column in Table 2 uses the same semi-supervised configurations with strengths $s_{G},s_{P}\in \left\{0,1\right\}$. Moreover, note that the case $\left(s_{G},s_{P}\right)=\left(0,0\right)$ is slightly different from supervised training in that batch normalization statistics are still updated in the student. The differences in results are due to variance—the estimated standard error of the mean of 5 runs is between 0.35 and 0.5. We can observe that the photometric component is more important, and that a stronger photometric component can compensate for a weaker geometric component. Our perturbation strength choice $\left(s_{G},s_{P}\right)=\left(1,1\right)$ is close to the optimum, which the experiments suggests to be at (1,0.5).

Figure 6 shows our validation of the consistency loss weight α with SN-RN18 simple-PhTPS. We observe the best generalization performance for $\alpha \in \left[0.25\;\ldots \;0.75\right]$. We do not scale the learning rate with ${\left(1+\alpha \right)}^{-1}$ because we use a scale-invariant optimization algorithm.

Appendix B presents experiments that quantify the effect of updating batch normalization statistics when the inputs are perturbed.

Figure 7 shows qualitative results on the first few validation images with SwiftNet-RN18 trained with 1/4 of labels. We observe that our method displays a substantial resilience to heavy perturbations, such as those used during training.

### 4.3. Semantic Segmentation on Full-Resolution Cityscapes

Table 3 presents our full resolution experiments in setups such as in Table 1, and comparison with previous work, but with full-resolution images and labels. In comparison with KE-GAN [53] and ECS [57], we underperform with 1/8 labeled images, but outperform with 1/2 labeled images. Note that KE-GAN [53] also trains on a large text corpus (MIT ConceptNet) as well as that ECS DLv3${}^{+}$-RN50 requires 22 GiB of GPU memory with batch size 6 [57], while our SN-RN18 simple-PhTPS requires less than 8 GiB of GPU memory with batch size 8 and can be trained on affordable GPU hardware. Section A.2 presents more detailed memory and execution time comparisons with other algorithms.

We note that the concurrent approach DLv3${}^{+}$-RN50 CAC [58] outperforms our method with 1/8 and 1/1 labels. However, ResNet-18 has significantly less capacity than ResNet-50. Therefore, the bottom section applies our method to the SwiftNet model with a ResNet-34 backbone, which still has less capacity than ResNet-50. The resulting model outperforms DLv3${}^{+}$-RN50 CAC across most configurations. This shows that our method consistently improves when more capacity is available.

We note that training DLv3${}^{+}$-RN50 CAC requires three RTX 2080Ti GPUs [58], while our SN-RN34 simple-PhTPS setup requires less than 9 GiB of GPU memory and fits on a single such GPU. Moreover, SN-RN34 has about 4× faster inference than DLv3${}^{+}$-RN50 on RTX 2080Ti.

Finally, we present experiments in the large-data regime, where we place the whole fine subset into $D_{l}$. In some of these experiments, we also train on the large coarsely labeled subset. We denote the extent of supervision with subscripts “l” (labeled) and “u” (unlabeled). Hence, $C_{u}$ in the table denotes the coarse subset without labels. Table 4 investigates the impact of the coarse subset on the SwiftNet performance on the full-resolution Cityscapes val. We observe that semi-supervised learning brings considerable improvement with respect to fully supervised learning on fine labels only (columns $F_{l}$ vs. $F_{l}\cup C_{u}$). It is also interesting to compare the proposed semi-supervised setup ($F_{l}\cup C_{u}$) with classic fully supervised learning on both subsets ($F_{l},C_{l}$). We observe that semi-supervised learning with SwiftNet-RN18 comes close to supervised learning with coarse labels. Moreover, semi-supervised learning prevails when we plug in the SwiftNet-RN34. These experiments suggest that semi-supervised training represents an attractive alternative to coarse labels and large annotation efforts.

### 4.4. Validation of Consistency Variants

Table 5 presents experiments with supervised baselines and four variants of semi-supervised consistency training. All semi-supervised experiments use the same PhTPS perturbations on CIFAR-10 (4000 labels and 50,000 images) and half-resolution Cityscapes (the SwiftNet-RN18 setups with 1/4 labels from Table 1). We investigate the following kinds of consistency: one-way with clean teacher (1w-ct, cf. Figure 1c), one-way with clean student (1w-cs, cf. Figure 1b), two-way with one clean input (2w-c1, cf. Figure 1d), and one-way with both inputs perturbed (1w-p2). Note that two-way consistency is not possible with Mean Teacher. Moreover, when both inputs are perturbed (1w-p2), we have to use the inverse geometric transformation on dense predictions [20]. We achieve that by forward warping [72] with the same displacement field. Two-way consistency with both inputs perturbed (2w-p2) is possible as well. We expect it to behave similarly to 1w-2p because it could be observed as a superposition of two opposite one-way consistencies, and our preliminary experiments suggest as much.

We observe that 1w-ct outperforms all other variants, while 2w-c1 performs in-between 1w-ct and 1w-cs. This confirms our hypothesis that predictions in clean inputs make better consistency targets. We note that 1w-p2 often outperforms 1w-cs, while always underperforming with respect to 1w-ct. A closer inspection suggests that 1w-p2 sometimes learns to cheat the consistency loss by outputting similar predictions for all perturbed images. This occurs more often when batch normalization uses the batch statistics estimated during training. A closer inspection of 1w-cs experiments on Cityscapes indicates the consistency cheating combined with severe overfitting to the training dataset.

### 4.5. Image Classification on CIFAR-10

Table 6 evaluates the image classification performance of two supervised baselines and 4 semi-supervised algorithms on CIFAR-10. The first supervised baseline uses only labeled data with standard data augmentation. The second baseline additionally uses our perturbations for data augmentation. The third algorithm is VAT with entropy minimization [4]. The simple-PhTPS approach outperforms supervised approaches and VAT. Again, two-way consistency results in the worst generalization performance. Perturbing the teacher input results in accuracy below 17% for 4000 or less labeled examples, and is not displayed. Note that somewhat better performance can be achieved by complementing consistency with other techniques that are either unsuitable for dense prediction or out of the scope of this paper [5,49,69].

## 5. Discussion

We have presented the first comprehensive study of one-way consistency for semi-supervised dense prediction, and proposed a novel perturbation model, which leads to the competitive generalization performance on Cityscapes. Our study clearly shows that one-way consistency with clean teacher outperforms other forms of consistency (e.g., clean student or two-way) both in terms of generalization performance and training footprint. We explain this by observing that predictions in perturbed images tend to be less reliable targets.

The proposed perturbation model is a composition of a photometric transformation and a geometric warp. These two kinds of perturbations have to be treated differently, since we desire invariance to the former and equivariance to the latter. Our perturbation model outperforms CutMix both in standard experiments with DeepLabv2-RN101 and in combination with recent efficient models (SwiftNet-RN18 and SwiftNet-RN34).

We consider two teacher formulations. In the simple formulation, the teacher is a frozen copy of the student. In the Mean Teacher formulation, the teacher is a moving average of student parameters. Mean Teacher outperforms simple consistency in low data regimes (half resolution; few labels). However, experiments with more data suggest that the simple one-way formulation scales significantly better.

To the best of our knowledge, this is the first account of semi-supervised semantic segmentation with efficient models. This combination is essential for many practical real-time applications where there is a lack of large datasets with suitable pixel-level groundtruth. Many of our experiments are based on SwiftNet-RN18, which behaves similarly to DeepLabv2-RN101, while offering about 9× faster inference on half-resolution images, and about 15× faster inference on full-resolution images on RTX 2080Ti. Experiments on Cityscapes coarse reveal that semi-supervised learning with one-way consistency can come close and exceed full supervision with coarse annotations. Simplicity, competitive performance and speed of training make this approach a very attractive baseline for evaluating future semi-supervised approaches in the dense prediction context.

## Figures and Tables

**Figure 1 sensors-23-00940-f001:**
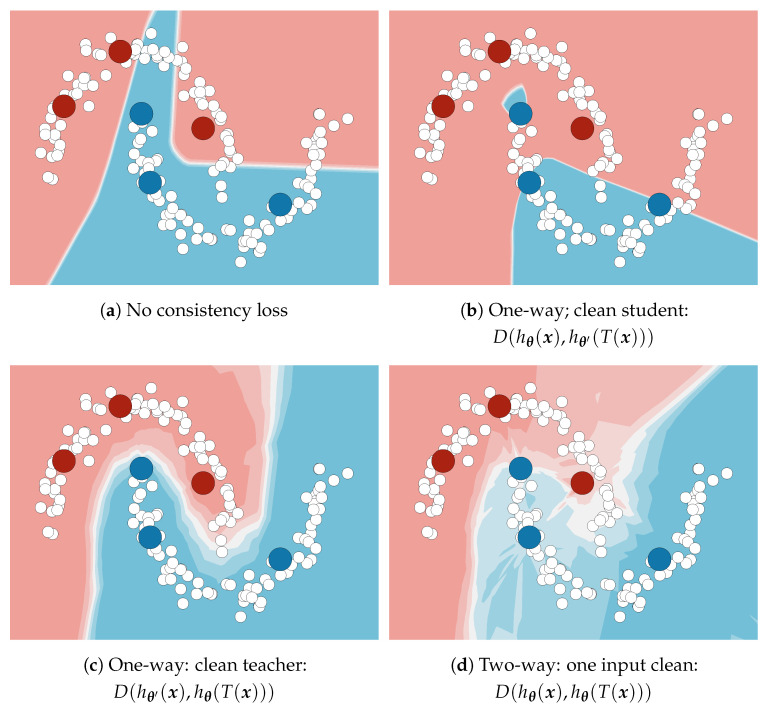
A toy semi-supervised classification problem with six labeled (red, blue) and many unlabeled 2D datapoints (white). All setups involve 20,000 epochs of semi-supervised training with cross-entropy and default Adam optimization hyper-parameters. The consistency loss was set to none (**a**), one-way with clean student (**b**), one-way with clean teacher (**c**), and two-way with one input clean (**d**). One-way consistency with clean teacher outperforms all other formulations.

**Figure 2 sensors-23-00940-f002:**
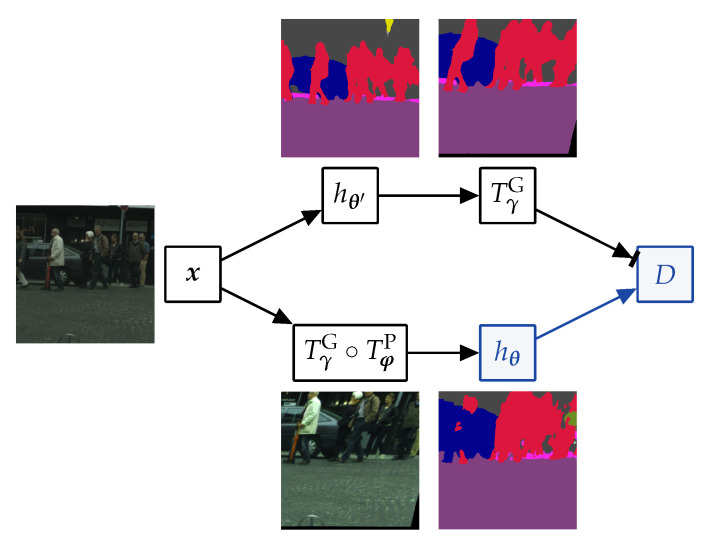
Dense one-way consistency with clean teacher. Top branch: the input is fed to the teacher $h_{\theta^{\prime }}$. The resulting predictions are perturbed with geometric perturbations $T_{\gamma }^{G}$. Bottom branch: the input is perturbed with geometric and photometric perturbations and fed to the student $h_{\theta }$. The loss *D* corresponds to the average pixel-wise KL divergence between the two branches. Gradients are computed only in the blue part of the graph.

**Figure 3 sensors-23-00940-f003:**
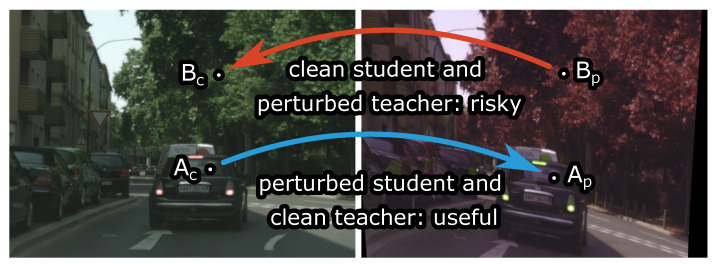
Two variants of one-way consistency training on a clean image (**left**) and its perturbed version (**right**). The arrows designate information flow from the teacher to the student. The proposed clean-teacher formulation trains in the perturbed pixels ($A_{p}$) according to the corresponding predictions in the clean image ($A_{c}$). The reverse formulation (training in $B_{c}$ according to the prediction in $B_{p}$) worsens performance, since strongly perturbed images often give rise to less accurate predictions.

**Figure 4 sensors-23-00940-f004:**
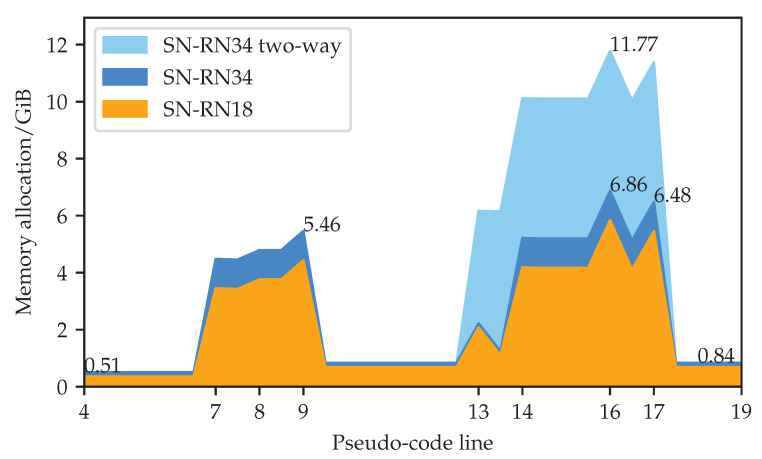
GPU memory allocation during and after execution of particular lines from Algorithm 1 during the 2nd iteration of training. Our PyTorch implementations involve SwiftNet-RN18 and SwiftNet-RN34 models with one-way and two-way consistency, 768×768 crops, and batch sizes $\left(B_{l},B_{u}\right)=\left(8,8\right)$. Line 9 computes the supervised gradient. Line 13 computes the teacher output (without caching interemediate activations). Lines 16 and 17 compute the consistency loss and its gradient.

**Figure 5 sensors-23-00940-f005:**
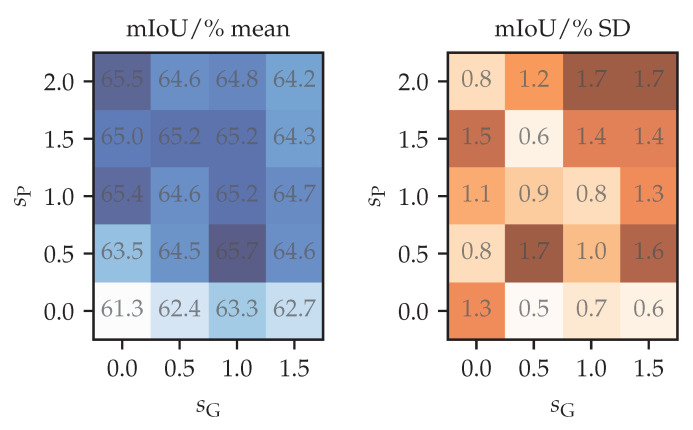
Validation of perturbation strength hyper-parameters on Cityscapes val (mIoU/%). We use 5 different subsets with 1/4 of the total number of training labels. The hyper-parameters $s_{P}$ (photometric) and $s_{G}$ (geometric) are defined in the main text. SD denotes the sample standard deviation.

**Figure 6 sensors-23-00940-f006:**
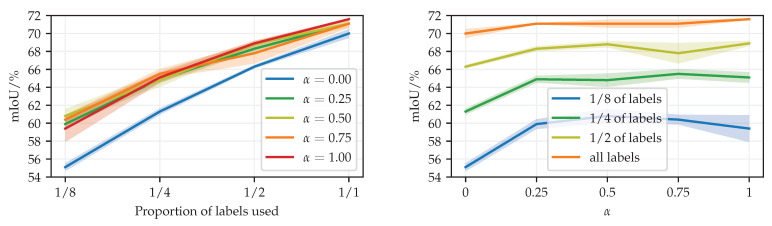
Validation of the consistency loss weight α on Cityscapes val (mIoU/%). We present the same results in two plots with different x-axes: the proportion of labels (**left**), and the consistency loss weight α (**right**).

**Figure 7 sensors-23-00940-f007:**
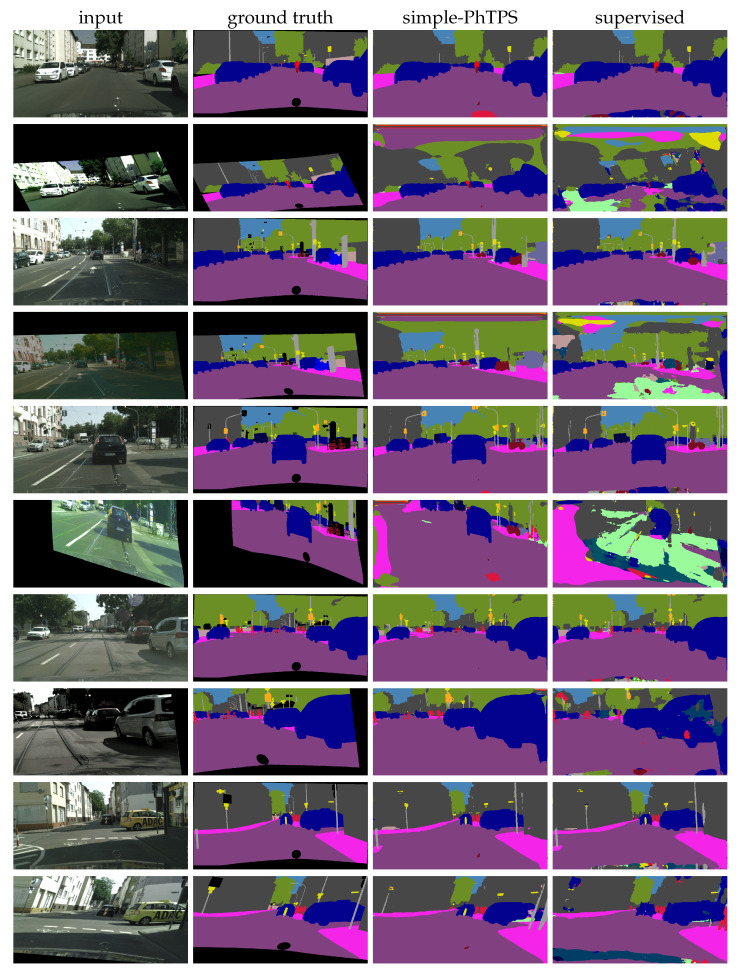
Qualitative results on the first few validation images with SwiftNet-RN18 trained with 1/4 of half-resolution Cityscapes labels. Odd rows contain unperturbed inputs, and even rows contain PhTPS perturbed inputs. The columns are (left to right): ground truth segmentations, predictions of simple-PhTPS consistency training, and predictions of supervised training.

**Table 1 sensors-23-00940-t001:** Semantic segmentation performance (mIoU/%) on half-resolution Cityscapes val after training with different proportions of labeled data. The top section reviews experiments from previous work. The middle section presents our experiments with DeepLab v2. The bottom section presents our experiments with SwiftNet-RN18. We run experiments across 5 different dataset splits and report mean mIoUs with standard deviations. The subscript “∼[25]” denotes training with $L^{2}$ loss, confidence thresholding, and α=1, as proposed in [25]. The best results overall are bold, and best results within sections are underlined.

Method	Label Proportion			
	1/8	1/4	1/2	1/1
DLv2-RN101 supervised [8,25]	56.20.0	60.20.0	64.60.01	66.00.0
DLv2-RN101 s4GAN+MLMT [8]	59.30.0	61.90.0	–	65.80.0
DLv2-RN101 supervised [7]	55.50.0	59.90.0	64.10.0	66.40.0
DLv2-RN101 AdvSemSeg [7]	58.80.0	62.30.0	65.70.0	67.70.0
DLv2-RN101 supervised [57]	56.00.0	60.50.0	–	66.00.0
DLv2-RN101 ECS [57]	60.3_0.0	63.8_0.0	–	67.7_0.0
DLv2-RN101 MT-CutMix [25]	${\underline{60.3}}_{1.2}$	${\underline{63.9}}_{0.7}$	–	${\underline{67.7}}_{0.4}$
DLv2-RN101 supervised	$56.4_{0.4}$	$61.9_{1.1}$	$66.6_{0.6}$	$69.8_{0.4}$
DLv2-RN101 MT-CutMix∼[25]	$63.2_{1.4}$	$65.6_{0.8}$	$67.6_{0.4}$	$70.0_{0.3}$
DLv2-RN101 MT-PhTPS	$61.5_{1.0}$	$66.4_{1.1}$	$69.0_{0.6}$	$71.0_{0.7}$
SN-RN18 supervised	$55.5_{0.9}$	$61.5_{0.5}$	$66.9_{0.7}$	$70.5_{0.6}$
SN-RN18 simple-CutMix	$59.8_{0.5}$	$63.8_{1.2}$	$67.0_{1.4}$	$69.3_{1.1}$
SN-RN18 simple-PhTPS	$60.8_{1.6}$	$64.8_{1.5}$	$68.8_{0.7}$	$71.1_{0.9}$
SN-RN18 MT-CutMix∼[25]	$61.6_{0.9}$	$64.6_{0.5}$	$67.6_{0.7}$	$69.9_{0.6}$
SN-RN18 MT-CutMix	$59.3_{1.3}$	$63.3_{1.0}$	$66.8_{0.6}$	$69.7_{0.5}$
SN-RN18 MT-PhTPS	${\underline{62.0}}_{1.3}$	$66.0_{1.0}$	$69.1_{0.5}$	$71.2_{0.7}$

**Table 2 sensors-23-00940-t002:** Ablation experiments on half-resolution Cityscapes val (mIoU/%) with SwiftNet-RN18. Subscripts denote the difference from the supervised baseline. The label “supervised PhTPS-aug” denotes supervised training where half of each mini-batch is perturbed with PhTPS. The bottom three rows compare PhTPS with Ph (only photometric) and TPS (only geometric) under simple one-way consistency. We present means of experiments on 5 different dataset splits. Numerical subscripts are differences with respect to the supervised baseline.

Method	Label Proportion			
	1/8	1/4	1/2	1/1
SN-RN18 supervised	55.5+0.0	61.5+0.0	66.9+0.0	70.5+0.0
SN-RN18 supervised PhTPS-aug	$56.2_{+1.5}$	$62.2_{+0.7}$	$67.4_{+0.5}$	$70.4_{-0.1}$
SN-RN18 simple-Ph	$59.2_{+3.7}$	$64.9_{+3.4}$	$68.3_{+1.4}$	$71.8_{+1.3}$
SN-RN18 simple-TPS	$58.2_{+3.1}$	$63.4_{+1.9}$	$66.7_{-0.2}$	$70.1_{-0.4}$
SN-RN18 simple-PhTPS	$60.8_{+5.3}$	$64.8_{+3.3}$	$68.8_{+1.9}$	$71.1_{+0.6}$

**Table 3 sensors-23-00940-t003:** Semi-supervised semantic segmentation performance (mIoU/%) on full-resolution Cityscapes val with different proportions of labeled data. We compare simple-PhTPS and MT-PhTPS (ours) with supervised training and previous work. DLv3${}^{+}$-RN50 stands for DeepLab v3${}^{+}$ with ResNet-50, and SN for SwiftNet. We run experiments across 5 different dataset splits and report mean mIoUs with standard deviations. Best results overall are bold, and best results within sections are underlined.

Method	Label Proportion			
	1/8	1/4	1/2	1/1
KE-GAN [53]	66.90.0	70.60.0	72.20.0	75.30.0
DLv3${}^{+}$-RN50 supervised [57]	63.20.0	68.40.0	72.90.0	74.80.0
DLv3${}^{+}$-RN50 ECS [57]	67.40.0	70.70.0	72.90.0	74.80.0
DLv3${}^{+}$-RN50 supervised [58]	63.90.0	68.30.0	71.20.0	76.30.0
DLv3${}^{+}$-RN50 CAC [58]	69.70.0	72.7_0.0	−0.0	77.5_0.0
SN-RN18 supervised	$61.1_{0.4}$	$67.3_{1.1}$	$71.9_{0.1}$	$75.4_{0.4}$
SN-RN18 simple-PhTPS	$66.3_{1.0}$	$71.0_{0.5}$	${\underline{74.3}}_{0.7}$	${\underline{75.8}}_{0.4}$
SN-RN18 MT-PhTPS	${\underline{68.6}}_{0.6}$	${\underline{72.0}}_{0.3}$	$73.8_{0.4}$	$75.0_{0.4}$
SN-RN34 supervised	$64.9_{0.8}$	$69.8_{1.0}$	$73.8_{1.4}$	$76.1_{0.8}$
SN-RN34 simple-PhTPS	$69.2_{0.8}$	$73.1_{0.7}$	$76.3_{0.7}$	$77.9_{0.2}$
SN-RN34 MT-PhTPS	$70.8_{1.5}$	$74.3_{0.5}$	$76.0_{0.5}$	$77.2_{0.4}$

**Table 4 sensors-23-00940-t004:** Effects of an additional large dataset on supervised and semi-supervised learning on full-resolution Cityscapes val (mIoU/%). Tags F and C denote fine and coarse subsets, respectively. Subset indices denote whether we train with labels (l) or one-way consistency (u).

Method	$F_{l}$	$\left(F_{l},F_{u}\right)$	$\left(F_{l},F_{u}\cup C_{u}\right)$	$\left(F_{l},C_{l}\right)$
SN-RN18 simple-PhTPS	$75.4_{0.4}$	$75.8_{0.4}$	$76.5_{0.3}$	$76.9_{0.3}$
SN-RN18 MT-PhTPS		$75.0_{0.4}$	$75.5_{0.3}$	
SN-RN34 simple-PhTPS	$76.1_{0.8}$	$77.9_{0.2}$	$78.5_{0.4}$	$77.7_{0.4}$

**Table 5 sensors-23-00940-t005:** Comparison of 4 consistency variants under PhTPS perturbations: one-way with clean teacher (1w-ct), one-way with clean student (1w-cs), two-way with one input clean (2w-c1), and one-way with both inputs perturbed (1w-p2). Algorithms are evaluated on CIFAR-10 test (accuracy/%) while training on 4000 out of 50,000 labels (CIFAR-10, 2/25) and half-resolution Cityscapes val (mIoU/%) while training on 1/4 of labels from Cityscapes train with SwiftNet-RN18 (CS-half, 1/4).

Dataset	Method	sup.	1w-ct	1w-cs	2w-c1	1w-p2
CIFAR-10, 4k	WRN-28-2 simple-PhTPS	$80.8_{0.4}$	$90.8_{0.3}$	$69.3_{4.2}$	$72.9_{2.6}$	$73.3_{7.0}$
CIFAR-10, 4k	WRN-28-2 MT-PhTPS	$80.8_{0.4}$	$90.8_{0.4}$	$80.5_{0.5}$	-	$73.4_{1.4}$
CS-half, 1/4	SN-RN18 simple-PhTPS	$61.5_{0.5}$	$65.3_{1.9}$	01.61.0	$16.7_{3.0}$	$61.6_{0.5}$
CS-half, 1/4	SN-RN18 MT-PhTPS	$61.5_{0.5}$	$66.0_{1.0}$	$61.5_{1.4}$	-	$62.0_{1.1}$

**Table 6 sensors-23-00940-t006:** Classification accuracy [%] on CIFAR-10 test with WRN-28-2. We compare two supervised approaches (top), VAT with entropy minimization [4] (middle), and two-way and one-way consistency with our perturbations (bottom three rows). We report means and standard deviations of 5 runs. The label “supervised PhTPS-aug” denotes the supervised training, where half of each mini-batch is perturbed with PhTPS.

Method	Number of Labeled Examples			
	250	1000	4000	50,000
supervised	$31.8_{0.6}$	$59.3_{1.4}$	$81.0_{0.2}$	94.7
supervised PhTPS-aug	$48.7_{0.9}$	$67.2_{0.5}$	$81.7_{0.2}$	95.0
VAT + entropy minimization	$41.0_{2.5}$	$73.2_{1.5}$	$84.2_{0.4}$	$90.5_{0.2}$
1w-cs simple-PhTPS	$27.7_{5.9}$	$51.7_{3.5}$	$69.3_{4.2}$	$91.6_{1.5}$
2w-c1 simple-PhTPS	$30.3_{1.8}$	$54.8_{1.5}$	$72.9_{2.6}$	$95.9_{0.2}$
1w-ct simple-PhTPS	$68.8_{5.4}$	$84.2_{0.4}$	$90.6_{0.4}$	$96.2_{0.2}$

## Data Availability

Data are available in a publicly accessible repository that does not issue DOIs and are provided by third parties. The datasets that we use are available at https://www.cityscapes-dataset.com/ and https://www.cs.toronto.edu/~kriz/cifar.html (accessed on 8 January 2023). Source code that enables reproducibility of our experiments is available at https://github.com/Ivan1248/semisup-seg-efficient.

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
