# Peer review of "Revisiting Consistency for Semi-Supervised Semantic Segmentation"

_sensors, 2023, doi:10.3390/s23020940_

Round 1

Reviewer 1 Report

Summary

This paper studied consistency loss computed from the outputs of two semantic segmentation models to achieve high model accuracy by a semi-supervised learning approach. In the semi-supervised learning approaches studied in this paper, models were expected to output similar pixel masks for clean and perturbed input. The paper argued that models trained with one-way consistency loss with a clean teacher outperformed other models trained with other semi-supervised learning approaches. Experiments with CityScapes dataset showed that the proposed training method resulted in better models across different model architectures. Additional analysis (CIFAR and CityScapes) on the variation of the proposed consistency loss confirmed that the proposed one-way, clean teacher was the best setting. The paper extended upon an existing conference paper by adding more experiment results and detailed analyses. 

Comments

The paper was well written with a clear message and empirical support. The central claim was similar to the authors' conference paper with additional analysis. However, the additional experiments and analyses provided some new insights into the proposed method. The authors should clearly specify (at the end of the introduction) what parts of the conference paper and what novel insights/claims.

The authors provided detailed methodology and experimental design for the experiments. I appreciated that the experiments were conducted with five repeated experiments. We can confidently conclude that the proposed method outperforms other methods. However, it was unclear which part of the experiment supported the efficiency claims. The authors should provide direct empirical results or discussion for the claim (memory footprint, run time, number of parameters, training time). 

Author Response

Answer to R1.1.

Thank you for your kind and constructive feedback! We have added more information to the introduction about the added value of the new experiments.

Answer to R1.2.

The new Figure 4 plots GPU memory usage at particular lines of the pseudo-code in Algorithm 1. One-way consistency enables memory saving in the line 13 of the pseudo-code, where teacher activations are not cached since one-way consistency freezes the teacher parameters.

The new tables from the appendix review hyper-parameter values as well as compare the number of model parameters and the speed of inference and training across all models from our experiments.

Reviewer 2 Report

Congratulations for the novel technique entitle "Perturbation Model"

Kindly check the English of introduction and Methodology section. 

Author Response

Answer to R2.1.

Thank you for your kind and constructive feedback!

The revised manuscript introduces a number of language-related improvements throughout the text.

Reviewer 3 Report

In this paper, the one-way consistency of semi-supervised dense prediction is comprehensively studied for the first time, and a novel perturbation model is proposed. The article research is interesting. But in my opinion, there are the following issues that need to be resolved.

1.    At the end of the introduction, I suggest adding a description of the structure of the article. For example, “in section 2 ..., in section 3...,in section 4...”

2.    There are many places in the article where the sentence expression is not formal enough. The author should modify the sentence of the article to make it more consistent with the expression of the journal. For example, "We study semi-supervised approaches... We consider ... " in lines 33 and 34 on page 1.

3.    In the discussion part, based on the one-way consistency of semi-supervised dense prediction, a new perturbation model is proposed. The author points out that the consistency of the model in some aspects is better than that in other forms (e.g., clean students or two-way consistency). However, in my opinion, this description does not express the advantages of the article. I suggest the author to add specific comparison with other articles. The comparison table can be used to highlight the advantages of this article. The form of comparison table can be inspired by the following articles: Memristor-Based Neural Network Circuit With Multimode Generalization and Differentiation on Pavlov Associative Memory, in IEEE Transactions on Cybernetics, 2022, doi: 10.1109/TCYB.2022.3200751. Memristor-Based Neural Network Circuit of Full-Function Pavlov Associative Memory With Time Delay and Variable Learning Rate," in IEEE Transactions on Cybernetics, vol. 50, no. 7, pp. 2935-2945, July 2020, doi: 10.1109/TCYB.2019.2951520.

4.    The format of references in articles should be consistent. For example, [5] and [6], and so on.

Author Response

Answer to R3.1.

Thank you for your kind and constructive feedback!

The revised manuscript describes the structure of the article at the end of the introduction.

Answer to R3.2.

The revised manuscript rewords some expressions from active voice and the first person plural into a more formal style. Such replacements were adopted wherever they preserved the clarity of our text.

Please note that Nature and Science journals express explicit preference for active voice [1,2]. Furthermore, some of the most cited articles from the Sensors journal (e.g. [3], [4]) prefer sentences in active voice. Article [3] has over 2000 citations on Google scholar.

[1] https://www.nature.com/nature-portfolio/for-authors/write
[2] https://www.science.org/content/article/writing-science-storys-thing
[3] https://www.mdpi.com/1424-8220/16/1/115
[4] https://www.mdpi.com/1424-8220/19/2/326

Answer to R3.3.

The new tables A1, A2, A3, A4, and A5 provide a comprehensive comparison of all involved models with respect to hyper-parameter values, optimizer configuration, computational complexity of training, number of model parameters, as well as computational complexity of inference. In order to avoid poor typesetting due to large sizes of these tables, we propose to place these tables in the appendix.

Answer to R3.4.

We thank the reviewer for pointing out that our references to Arxiv preprints have been typeset acording to two different conventions. The revised manuscript reduces the number of preprint references to only three and typesets them consistently according to DBLP bibtex references.

Round 2

Reviewer 3 Report

Ok, it is OK now!